# A Comparison of Model Averaging Techniques to Predict the Spatial Distribution of Soil Properties

**Ruhollah Taghizadeh-Mehrjardi** [1,2,3], **Hossein Khademi** [4], **Fatemeh Khayamim** [4,*], **Mojtaba Zeraatpisheh** [5,6], **Brandon Heung** [7] **and Thomas Scholten** [1,2,3]

1   Department of Geosciences, Soil Science and Geomorphology, University of Tübingen, 72070 Tübingen, Germany; taghizadeh-mehrjardi@mnf.uni-tuebingen.de (R.T.-M.); thomas.scholten@uni-tuebingen.de (T.S.)
2   CRC 1070 Resource Cultures, University of Tübingen, 72074 Tübingen, Germany
3   DFG Cluster of Excellence "Machine Learning", University of Tübingen, 72070 Tübingen, Germany
4   Department of Soil Science, College of Agriculture, Isfahan University of Technology, Isfahan 8415683111, Iran; hkhademi@cc.iut.ac.ir
5   Henan Key Laboratory of Earth System Observation and Modeling, Henan University, Kaifeng 475004, China; mojtaba.zeraatpisheh@henu.edu.cn
6   College of Geography and Environmental Science, Henan University, Kaifeng 475004, China
7   Department of Plant, Food, and Environmental Sciences, Faculty of Agriculture, Dalhousie University, Truro, NS B2N 5E3, Canada; brandon.heung@dal.ca
*   Correspondence: f.khayamim@alumni.iut.ac.ir

**Abstract:** This study tested and evaluated a suite of nine individual base learners and seven model averaging techniques for predicting the spatial distribution of soil properties in central Iran. Based on the nested-cross validation approach, the results showed that the artificial neural network and Random Forest base learners were the most effective in predicting soil organic matter and electrical conductivity, respectively. However, all seven model averaging techniques performed better than the base learners. For example, the Granger–Ramanathan averaging approach resulted in the highest prediction accuracy for soil organic matter, while the Bayesian model averaging approach was most effective in predicting sand content. These results indicate that the model averaging approaches could improve the predictive accuracy for soil properties. The resulting maps, produced at a 30 m spatial resolution, can be used as valuable baseline information for managing environmental resources more effectively.

**Keywords:** spatial modeling; machine learning; remote sensing; model averaging





## 1. Introduction

In recent years, rapid population growth and the increasing demand for food have had undesirable consequences on the environment. These consequences include, but are not limited to, land degradation, desertification, water pollution, and soil pollution. Therefore, there is a need to explore and recognize the factors related to sustainable agriculture and soil and water resources management. Hence, one of the most basic pieces of information related to land resource management includes maps of soil properties [1]. Soil properties vary both temporally and spatially and from small- to large-scale, and are affected by environmental characteristics, such as topography, and soil management practices, such as fertilization and agronomic practices [2].

In Iran, where 85% of the country is arid or semiarid [3], the intrinsic properties of soil, such as SOM, CCE, gypsum content, soil texture, electrical conductivity (EC), soil pH, and soil reactivity have been shown to be related to soil quality and are commonly considered the main factors in soil quality assessments [4]. However, these properties are highly variable in space and time [5]—especially in agricultural systems, due to the

processes related to soil redistribution and agriculture practices. Lastly, parts of the region are challenged by data scarcity where soil information lacks detail or is not available.

Because soil information is essential, digital soil mapping (DSM) has been an area of research over the past few decades [1]. Whereas traditional soil mapping methods were time-consuming and expensive to carry out [6], DSM techniques can overcome these limitations by integrating soil information and environmental variables obtained from remote sensing and other geospatial datasets [1,7,8]. DSM approaches operate by establishing correlations between a set of environmental covariates and soil properties of georeferenced sample points in the study area. The resulting predictive models are then applied to unsampled locations. Until recently, most DSM studies have been carried out in easily accessible regions in Iran to predict a variety of soil properties; for instance, pH, EC, soil organic matter (SOM), phosphorus, particle size distribution, and calcium carbonate equivalent (CCE) [8–10]. However, few studies have investigated the spatial soil properties in regions with limited soil data or in difficult-to-access areas.

Machine learning (ML) techniques have increasingly been compared for identifying the best performing model for predicting soil variability [11]. Of the many ML algorithms currently used in DSM, studies have included the use of multiple linear regression [12], logistic regression [13], Random Forests [14–17], classification trees [18], support vector machines [17,19], and artificial neural networks [20]. However, with increasing computational power, more sophisticated and complex algorithms, such as convolutional neural networks, which are based on data-hungry, deep learning approaches, have been used to solve highly complex soil-landscape problems and to improve the prediction accuracy and decrease the uncertainty of digital soil maps [21–24].

An approach to improving the predictive capability and decreasing the variance of ML models is through model averaging [25]. Model averaging is a technique where multiple individual learners (i.e., base learners) are trained and combined to solve the same problem. This technique assumes that each base learner will have its own strengths and weaknesses and compile a final model with the strengths of the individual models. As a result, model averaging techniques are expected to produce predictions with similar or better accuracy when compared to their individual constituents. In addition to the increased accuracy, model averaging has the potential to improve the reliability, stability, and robustness of models [26]. These techniques have recently gained attention in environmental sciences, atmospheric sciences, and statistics literature for predicting and solving highly complex problems [21].

A few DSM studies have demonstrated the effectiveness of model averaging for predicting various soil properties, such as available soil water, soil organic carbon, soil texture, and soil pH [7]; hydrologic properties [25]; and soil classes [27]. However, to the best of our knowledge, there are no DSM studies that have performed a comprehensive comparison of model averaging techniques; hence, providing the impetus for this study.

Given the need for detailed soil information for the arid, remote, and data-scarce regions of Iran, this study aimed to compare and evaluate methods for producing maps of soil properties using ML and model averaging techniques. The specific objectives were as follows: (1) to investigate and compare the use of different single-model learners, such as support vector regression (SVR), k-nearest neighbor (kNN), artificial neural network (ANN), deep neural network (DNN), Random Forest (RF), adaptive network-based fuzzy inference system (ANFIS), and extreme gradient boosting (XGB); and (2) to compare the single-model learners with several model averaging techniques, such as Bates–Granger averaging (BGA), equal weights averaging (XBEWA), Bayesian information criterion (BIC), Akaike's information criterion (AIC), Bayesian model averaging (BMA), Granger–Ramanathan averaging (GRA), and Mallows model averaging (MMA).

## 2. Materials and Methods

### 2.1. Study Area and Soil Sampling

The research area is 110,000 km$^2$ and is located in the central Iranian province of Isfahan (Figure 1). The elevation varies from 700 to 2600 m above mean sea level. The mean annual precipitation and temperature are 117 mm and 25 °C, respectively. According to the geology map, quaternary sediments cover a considerable portion of the Isfahan province. Sedimentary rocks, such as limestones, sandstones, conglomerate, and shale, are common in the southern and western regions of the study area [28].

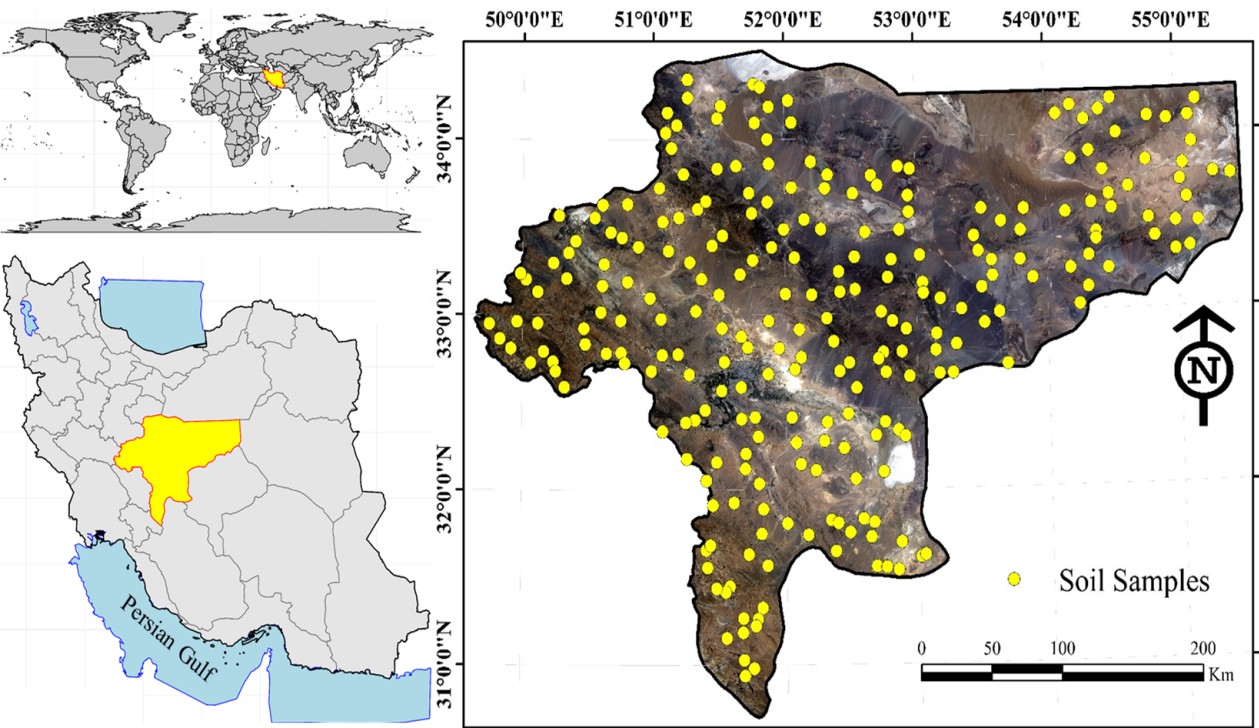

**Figure 1.** The location of Iran and the Isfahan province (**left**) and the spatial distribution of sampling points (**right**).

A total of 251 topsoil samples were collected at 0–20 cm depth increments, utilizing a stratified random sampling with 20 × 20 km stratification blocks. Soil samples within the blocks were selected based on parent material by taking one sample from the dominant parent material within each grid cell. This ensured that sedimentary, volcanic, and meta-morphic rocks were represented. Each sample consisted of five subsamples, which were randomly collected from within a 20 × 20 m (400 m$^2$) area in each grid. The geographical distribution of sample locations within the study area is shown in Figure 1. Soil samples were air-dried and sieved using a 2 mm sieve. Using a 2:1 water to soil ratio extract, the soil pH [29] and electrical conductivity (EC) [30] were measured. In addition, the SOM content (wet combustion method) [31], CCE (titration method) [32], particle size distri-bution (hydrometer method) [33], the gypsum content (oven-drying method) were also measured [34].

### 2.2. Environmental Covariates

A digital elevation model (DEM; Figure 2) with a 30 m spatial resolution was used to calculate terrain attributes [35], such as elevation, catchment aspect, catchment slope, catchment area, topographic openness, profile curvature, topographic wetness index, and planform curvature (Table 1). Additionally, Landsat 8 Operational Land Imager (OLI) data were taken during the summer of 2012. After performing geometric and radiometric

corrections on the Landsat images, the median values of bands were used to derive a suite of remote sensing covariates, such as brightness index, salinity index, gypsum index, carbonate index, and clay index (Table 1). Lastly, the mean annual temperature and mean annual precipitation (Figure 2) were calculated from the monthly precipitation and temperature values.

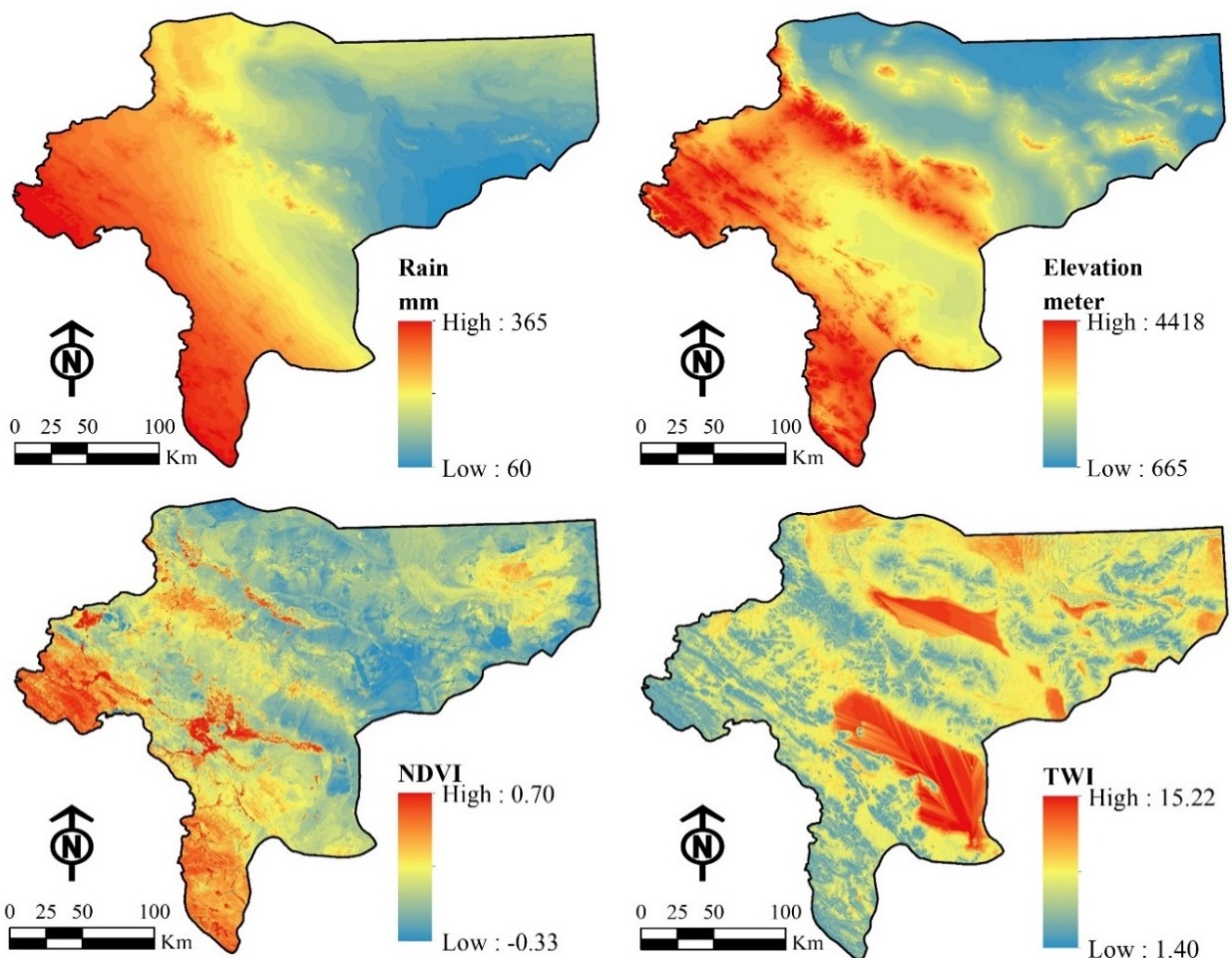

**Figure 2.** Examples of covariates for the Isfahan province.

**Table 1.** List of environmental covariates used (*L is a canopy background adjustment factor).

| Covariates | Definition | Code | Source and Ref. |
|---|---|---|---|
| Elevation | | X01_Elev | |
| Catchment Aspect | | X02_Catch.Asp | |
| Catchment Slope | | X03_Catch.Slop | |
| Catchments area | | X04_Catch.Area | |
| Openness (PosOpen) | | X05_Openness | |
| Profile curvature | | X06_Prf.Curv | DEM SRTM |
| Plan curvature | | X07_Pl.Curv | |
| Wetness index | | X08_Wetness.In | |
| Valley depth | | X09_Valley.Dep | |
| Slope length | | X11_Slop.Leng | |
| Total insolation | | X12_Total.Inso | |
| Multi-resolution valley bottom flatness index | | X10_MrVBF | DEM SRTM, [36] |

**Table 1.** *Cont.*

| Covariates | Definition | Code | Source and Ref. |
|---|---|---|---|
| Blue | B2: 0.45–0.51 μm | X13_B1 | |
| Green | B3: 0.53–0.59 μm | X14_B2 | |
| Red | B4: 0.64–0.67 μm | X15_B3 | Landsat 8, [37] |
| Near-infrared | B5: 0.85–0.88 μm | X16_B4 | |
| Short-wave infrared-1 | B6: 1.57–1.65 μm | X17_B5 | |
| Short-wave infrared-2 | B7: 2.11–2.29 μm | X18_B7 | |
| TASSELED CAP 1 | The overall brightness of the image | X19_TSC 1 | |
| TASSELED CAP 2 | The overall greenness of the image | X20_TSC 2 | Landsat 8, [38] |
| TASSELED CAP 3 | The overall wetness of the image | X21_TSC 3 | |
| Salinity index | $(B1 - B3)/(B1 + B3)$ | X22_Salinity.In | Landsat 8, [39] |
| Brightness index | $((B3)^2 + (B4)^2)^{0.5}$ | X23_Bright.In | Landsat 8, [40] |
| Gypsum index | $(B5 - B4)/(B5 + B4)$ | X24_Gypsum.In | |
| Clay index | B5/B7 | X25_Clay.In | Landsat 8, [41] |
| Carbonate index | B3/B2 | X26_Carbon.In | |
| Ratio vegetation index | $(B4/B3)/(B2 + B3)$ | X27_RVI | |
| Enhanced vegetation index | $(B4 - B3)/(B4 + C1 \times B3 - C2 \times B1 + L)$ | X28_EVI | |
| Infrared percentage vegetation index | $B4/(B4 + B3)$ | X29_IPVI | Landsat 8, [42] |
| Normalized difference vegetation index | $(B4 - B3)/(B4 + B3)$ | X30_NDVI | |
| Soil adjusted vegetation index | $*(1 + L) \times (B4 - B3)/(B4 + B3 + L)$ | X31_SAVI | |
| Annual mean temperature | It is derived from the monthly temperature values | X32_Temp | WordClim, [43] |
| Annual mean precipitation | It is derived from the monthly rainfall values | X33_Rainfall | |

### 2.3. Variable Importance Analysis Using a Genetic Algorithm

Many environmental covariates are often employed in DSM studies, thus making it difficult to understand the correlations between soils and the environment due to a large number of covariates. Brungard et al. [44] recommended that applying fewer covariates could benefit and improve the efficiency of the model process. To overcome this issue, genetic algorithms (GA) have been used to determine the optimal subsets of covariates to create simpler models while maintaining model performance [9,45]. Genetic algorithms are biologically inspired computational models based on evolutionary processes, such as selection, crossover, and mutation, and are designed to search for functions that best fit the experimental data set [45]. Here, a GA was applied to select the most important variables for each soil property using the *caret* package (version 6.0–90) in R [46].

### 2.4. Base Learners

Nine base learners were tested in establishing the relationships between the environmental covariates and target variables. These models included k-nearest neighbor (kNN), genetic programming (GP), support vector regression (SVR), least absolute shrinkage and selection operator (LASSO), artificial neural network (ANN), deep neural network (DNN), Random Forest (RF), adaptive network-based fuzzy inference system (ANFIS), extreme gradient boosting (XGB). Modeling was implemented using the *caret* package (version 6.0–90) [47] in R 3.2.5 [46] and RStudio (version 0.99.903) [48].

### 2.5. Model Averaging Techniques

Seven model averaging techniques were tested: Akaike's information criterion, equal weights averaging, Bates–Granger averaging, Bayes' information criterion, Mallows model averaging, Granger–Ramanathan averaging, and Bayesian model averaging. Here, we summarize each approach and refer readers to the references for full descriptions of each model averaging technique.

In the equal weights averaging (EWA) method, the final prediction is obtained by assigning the same weight to each model. In effect, this would be the mean predicted value amongst all base learners.

Bates–Granger averaging (BGA) technique was proposed by Bates and Granger [49]. In the BGA technique, each model is weighed by $1/\sigma_i{}^2$, where $\sigma_i{}^2$ is the prediction variance.

In the information criterion averaging techniques (AIC and BIC), weights are calculated using the following equations [50]:

$$\hat{\beta} = \frac{exp\left(\frac{I_i}{2}\right)}{\sum_{j=1}^{k} exp\left(\frac{I_i}{2}\right)} \tag{1}$$

where $I_i$ is an information criterion (the fit of the model), where

$$I_i = -2log(L_i) + q(p_i) \tag{2}$$

and $L_i$ is the (maximized) likelihood of model $i$, and $q(p_i)$ is a penalty for increasing the number of parameters, $p_i$, which needs to be estimated for model $i$. In the AIC averaging technique, the penalty, $q(p)$, is $2p$, while the penalty for the BIC averaging technique is $q(p) = p\log(n)$, where $n$ is the training sample size.

Hoeting et al. [51] first proposed the Bayesian model averaging (BMA) technique, which assigns a conditional probability density function to each model prediction. Raftery et al. [52] provide an excellent overview of the theoretical background behind the different BMA techniques.

Claeskens and Hjort [53] and Hjort and Claeskens [54] proposed the Mallows model averaging (MMA) technique and concluded that there is no best model; instead, an appropriate model should depend on the objective. In the following equation:

$$C_n(\beta) = \sum_{t=1}^{n} \left(Y_t - \beta' X_t\right)^2 + 2 \sum_{j=1}^{k} \beta_j p_j S^2 \tag{3}$$

$pj$ is the number of parameters of model $j$, and $S^2$ is an estimate of the variance, $\sigma^2$, of $\varepsilon_t$. In this study, $S^2$ was taken to be the smallest observed RMSE for any individual model among the set of models.

Granger and Ramanathan [55] first proposed the Granger–Ramanathan (GR) approach. It assumes that the final prediction is calculated from a combination of different model predictions using an ordinary least squares method.

*2.6. Accuracy Assessment and Uncertainty Analysis*

The dataset was randomly split into 70% ($n$ = 170) and 30% ($n$ = 81) for model training and testing, respectively. Leave-one-out cross-validation was also used to tune the hyperparameters of models using the 70% training dataset. The coefficient of determination ($R^2$), mean absolute error (MAE), the root mean squared error (RMSE), and the normalized root mean squared error (nRMSE) were used to assess model performance:

$$R^2 = 1 - \frac{\sum_{i=1}^{n}(y_i - \hat{y}_i)^2}{\sum_{i=1}^{n}(y_i - \overline{y}_i)^2} \tag{4}$$

$$RMSE = \sqrt{\frac{1}{n} \sum_{i=1}^{n}(y_i - \hat{y}_i)^2} \tag{5}$$

$$MAE = \frac{\sum_{i=1}^{n}|\hat{y}_i - y_i|}{n} \tag{6}$$

$$nRMSE = \frac{RMSE}{\overline{X}} \tag{7}$$

where in Equations (4)–(7), $y$ is the measured value, $\hat{y}$ is the predicted value, $n$ is the number of observations, and $\overline{X}$ is the average of observed values.

To assess the uncertainty of the models, a leave-one-out cross-validation method was used. This method resulted in 170 predicted soil property maps. Based on the predicted maps, the mean and standard deviation (SD) of soil properties for each pixel were calculated. Given a confidence level of 90%, the upper and lower boundary of the predictions (i.e., prediction interval) were calculated (mean $\pm$ 1.64 SD). Finally, the proportion of measured soil values that fell within the 90% prediction interval (i.e., prediction interval coverage probability; PICP) and mean prediction interval (MPI: upper prediction limit minus the lower prediction limit) were calculated as two measures of the quality of the uncertainty estimates.

## 3. Results and Discussion

### 3.1. Descriptive Statistics of Soil Properties

Descriptive statistics of soil properties are presented in Table 2. The SOM, CCE, gypsum, silt, sand, and EC values varied widely in the study area; for example, the CCE ranged from 0.2% to 80.0%, with a mean value of 27.8%. Due to the limestone-enriched parent materials, most soils are highly calcareous throughout the region [28], and because of the low precipitation in arid and semiarid regions, calcium carbonates tend to accumulate in the surficial soils [56]. SOM was low with a mean value of 0.4 %, which was also attributed to the arid and semiarid climate of the study area. Gypsum, sand, and EC values ranged substantially; however, the mean gypsum and EC remained low. Regions with high gypsum, sand, and EC values were located in the arid parts of the study area with low precipitation and high temperatures. The SOM, gypsum, and EC values were positively skewed, whereas the lime, clay silt, sand, and pH values followed a normal distribution (Table 2).

**Table 2.** Summary statistics of soil properties.

| Parameter | Number | Unit | Minimum | Maximum | Mean | SD | Skewness | Kurtosis |
|-----------|--------|------|---------|---------|------|------|----------|----------|
| SOM | 251 | % | 0.0 | 2.5 | 0.4 | 0.5 | 1.9 | 3.6 |
| CCE | 251 | % | 0.2 | 80.0 | 27.8 | 17.7 | 0.5 | −0.6 |
| Gypsum | 251 | % | 0.0 | 61.7 | 5.4 | 7.7 | 3.4 | 16.0 |
| Clay | 251 | % | 2.0 | 38.8 | 12.9 | 7.5 | 0.7 | −0.1 |
| Silt | 251 | % | 2.0 | 85.0 | 31.4 | 16.9 | 0.7 | 0.3 |
| Sand | 251 | % | 0.2 | 94.7 | 55.6 | 20.2 | −0.3 | −0.7 |
| EC | 251 | dS/m | 0.1 | 78.7 | 3.3 | 9.0 | 4.8 | 28.1 |
| pH | 251 | $-\log(H^+)$ | 7.1 | 8.7 | 7.9 | 0.2 | 0.2 | 1.1 |

### 3.2. Variable Importance Analysis

As illustrated in Figure 3, temperature, elevation, rainfall, NDVI, and SAVI indices were the most important covariates in predicting SOM content. Compared to other studies, Zeraatpisheh et al. [8] indicated that RVI, elevation, and SAVI were the most important covariates for SOM prediction in Iran. In contrast, Wang et al. [57] and Ayoubi et al. [58] concluded that topographic attributes significantly influenced SOM due to its effects on runoff, drainage, and soil erosion. Additionally, several studies in Iran demonstrated a strong relationship between vegetation cover and soil properties whereby vegetation indices were effective in capturing the variability in soil properties, especially SOM [59].

The most important predictors of silt content were the MRVBF, clay, and brightness indices. In comparison, the prediction of sand contents relied on rainfall, clay index, and elevation, whereas clay predictions were more reliant on rainfall, temperature, and elevation were the most important covariates (Figure 3). Thus, in the study area, climatic factors such as rainfall and temperature, along with topography attributes, could reflect the soil redistribution process due to water and wind [60]. For example, Brierley et al. [61] reported that in inter-rill soil erosion, the selective removal process led to the redistribution of silt and clay particles. Mosleh et al. [62] indicated that effective predictors of silt variability in Iran included diffuse radiation and wetness index, while the most important predictors of

clay content were aspect, duration of solar radiation, and stream power index (SPI). They believed that this was possibly related to the covariates being able to better represent the effects of vertical and lateral movements of soil particles through erosion and deposition processes in their study area. The importance of topographic predictors in mapping particle size fractions in Iran has also been demonstrated in studies, such as Zeraatpisheh et al. [8], which reported that curvature and profile curvature were important controls on water flow in the landscape and thereby explained most of the spatial distribution of clay content. Elsewhere, Adhikari et al. [63] showed that land use, soil, and landscape types were more important in predicting silt and also indicated that the distribution of fine and coarse sand fractions was effectively predicted by slope, elevation, and geology in Denmark. Nath [64] reported that the stream power index and topographic wetness index were the key predictors of sand content in the Northwest Iowa plain. In Nigeria, Akpa et al. [65] demonstrated that topographic variables (e.g., SPI, elevation, and slope), vegetative indices, and climatic variables were the most important predictors of soil particle size.

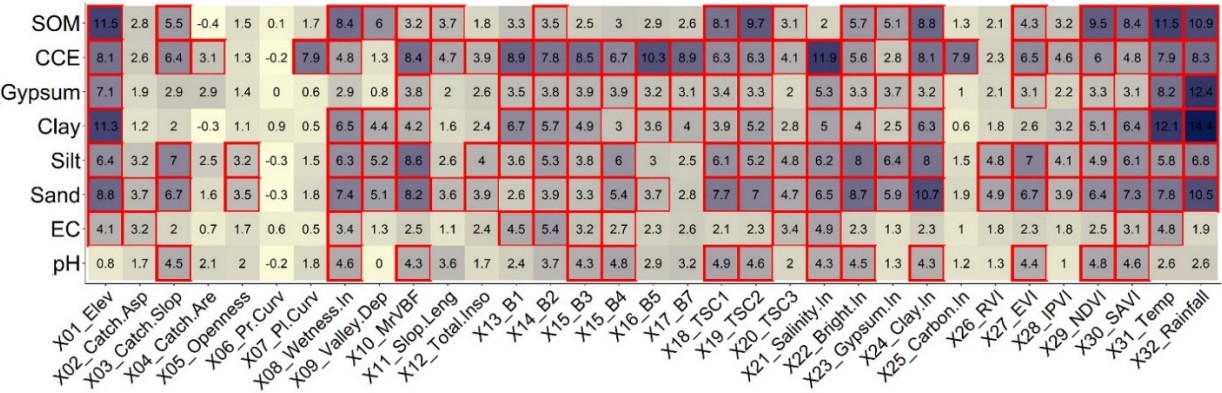

**Figure 3.** Importance of covariates for soil properties.

The prediction of gypsum was controlled mainly by rainfall, temperature, and elevation, while CCE predictions were controlled by salinity index as well as Bands 5 and 7 from the Landsat 8 data (Figure 3). Perhaps, this was due to the effect of climate and the different solubility rates of gypsum and calcium carbonates, where the lower solubility rate of calcium carbonates resulted in its presence on the soil surface, thus making it more visible in satellite imagery [66]. When predicting EC, the most influential predictors were temperature, Band 2, and salinity index (Figure 3), which was in contrast to Mosleh et al. [62], where they reported that elevation, curvature, planform curvature, and profile curvature were the key predictors of EC. For pH predictions, the Tasseled Cap Bands 1, 2, and NDVI were the most important predictor, which was in contrast to Mosleh et al. [62] and Nath [64], which reported the importance of planform curvature.

Overall, climatic parameters (e.g., rainfall and temperature), elevation, and RS data were the most important covariates for predicting the soil properties of the region. For CCE, EC, and pH, remote sensing data was particularly effective due to the accumulation of salts at the surface of the soil, which was easily detected by RS imagery. Meier et al. [67] selected 10 covariates for soil mapping, including four topographic covariates, three images from Landsat, two climatic maps, and the map of Euclidean distance from the drainage network. This study showed that MRVBF, temperature, rainfall, and TWI were the most important covariates for soil mapping (Figure 3). Mosleh et al. [62] concluded that terrain attributes were the main predictors for predicting soil properties, while other studies demonstrated the importance of remotely sensed vegetation parameters in the semiarid regions of Iran [9,17].

### 3.3. Comparison of Base Learners

Among the eight soil properties, the ANN model performed the best in predicting SOM, CCE, and gypsum content, while the RF model performed the best in predicting EC and pH. The best performing model for particle size classes varied, where ANFIS, XGB, and DNN were the most effective in predicting the sand, silt, and clay fractions, respectively (Table 3). Khaledian and Miller [68] concluded that the ANN model would likely produce the best results for large datasets, although the computational time could drastically increase compared to the other models, such as RF and SVR. However, in this study, the efficiency of the computational process was not a serious issue due to the limited size of our dataset. Mosleh et al. [62] and Were et al. [69] also found that the ANN model showed superior performance in predicting SOM compared to others. For predicting sand contents, similar results were reported in Taghizadeh-Mehrjardi et al. [70], where it was found that the ANFIS model had better performance when compared to multiple linear regression and ANN. Although kNN learner was ineffective in predicting soil properties (Table 3), other studies have demonstrated its effectiveness, such as Khaledian and Miller [68]. It is difficult to explain the reasons for these differences; however, the differences could be related to the different extents of the study areas, topography, sampling densities, or the quantity and quality of the environmental covariates used. Furthermore, this suggests that there is no single 'best' ML algorithm and that multiple models should be compared to identify the most appropriate model.

**Table 3.** Summary of accuracy metrics for base learners and model averaging techniques.

| Soil Property | Validation | Base Learner | | | | | | | | | Model Averaging Technique | | | | | | |
|---|---|---|---|---|---|---|---|---|---|---|---|---|---|---|---|---|---|
| | | kNN | SVR | GP | Lasso | ANN | DNN | ANFIS | RF | XGB | EWA | BGA | AIC | BIC | BMA | MMA | GRA |
| SOM | RMSE | 0.32 | 0.29 | 0.28 | 0.27 | 0.24 | 0.26 | 0.26 | 0.26 | 0.26 | 0.25 | 0.25 | 0.25 | 0.25 | 0.25 | 0.25 | 0.23 |
| | $R^2$ | 0.69 | 0.74 | 0.73 | 0.76 | 0.77 | 0.75 | 0.77 | 0.77 | 0.75 | 0.77 | 0.76 | 0.77 | 0.82 | 0.75 | 0.81 | 0.84 |
| | MAE | 0.22 | 0.18 | 0.17 | 0.17 | 0.16 | 0.18 | 0.17 | 0.17 | 0.18 | 0.16 | 0.18 | 0.17 | 0.17 | 0.18 | 0.17 | 0.15 |
| | nRMSE | 0.68 | 0.62 | 0.59 | 0.58 | 0.52 | 0.56 | 0.56 | 0.55 | 0.56 | 0.54 | 0.54 | 0.54 | 0.54 | 0.54 | 0.53 | 0.51 |
| CCE | RMSE | 12.19 | 12.14 | 14.77 | 11.82 | 11.48 | 12.14 | 11.91 | 11.84 | 11.98 | 11.80 | 12.00 | 11.49 | 11.79 | 11.45 | 11.46 | 11.42 |
| | $R^2$ | 0.55 | 0.55 | 0.34 | 0.56 | 0.59 | 0.55 | 0.55 | 0.58 | 0.57 | 0.57 | 0.58 | 0.63 | 0.59 | 0.62 | 0.63 | 0.65 |
| | MAE | 9.42 | 9.24 | 11.27 | 8.96 | 8.64 | 9.63 | 9.20 | 9.21 | 9.42 | 8.88 | 9.37 | 8.98 | 9.48 | 8.85 | 9.33 | 8.24 |
| | nRMSE | 0.44 | 0.44 | 0.53 | 0.42 | 0.41 | 0.44 | 0.43 | 0.43 | 0.43 | 0.42 | 0.43 | 0.41 | 0.42 | 0.41 | 0.41 | 0.41 |
| Gypsum | RMSE | 6.14 | 6.27 | 7.14 | 6.48 | 6.10 | 6.16 | 6.33 | 6.47 | 5.95 | 5.68 | 5.67 | 5.73 | 5.42 | 5.63 | 5.44 | 5.63 |
| | $R^2$ | 0.37 | 0.34 | 0.19 | 0.37 | 0.43 | 0.40 | 0.37 | 0.39 | 0.44 | 0.44 | 0.47 | 0.46 | 0.54 | 0.50 | 0.52 | 0.45 |
| | MAE | 3.76 | 3.57 | 4.19 | 3.61 | 3.50 | 3.50 | 3.86 | 3.88 | 3.92 | 3.41 | 3.72 | 3.56 | 3.78 | 3.42 | 3.50 | 3.69 |
| | nRMSE | 1.11 | 1.13 | 1.29 | 1.17 | 1.10 | 1.11 | 1.14 | 1.17 | 1.08 | 1.03 | 1.03 | 1.02 | 1.03 | 0.98 | 1.02 | 1.02 |
| Sand | RMSE | 15.24 | 15.09 | 16.34 | 15.50 | 14.86 | 14.98 | 14.66 | 14.97 | 15.00 | 14.83 | 15.00 | 15.18 | 14.84 | 14.38 | 14.73 | 14.87 |
| | $R^2$ | 0.46 | 0.45 | 0.39 | 0.44 | 0.48 | 0.46 | 0.47 | 0.39 | 0.48 | 0.46 | 0.48 | 0.47 | 0.51 | 0.56 | 0.55 | 0.54 |
| | MAE | 12.04 | 11.72 | 12.95 | 12.41 | 11.65 | 12.11 | 12.04 | 12.70 | 12.28 | 11.91 | 12.21 | 12.30 | 12.31 | 11.93 | 12.08 | 12.02 |
| | nRMSE | 0.27 | 0.27 | 0.29 | 0.28 | 0.27 | 0.27 | 0.27 | 0.27 | 0.27 | 0.27 | 0.27 | 0.27 | 0.27 | 0.26 | 0.27 | 0.27 |
| Clay | RMSE | 6.08 | 6.17 | 6.18 | 5.99 | 5.98 | 5.94 | 5.98 | 5.96 | 5.92 | 5.87 | 5.92 | 5.80 | 5.93 | 5.68 | 5.83 | 5.96 |
| | $R^2$ | 0.38 | 0.34 | 0.39 | 0.40 | 0.42 | 0.40 | 0.46 | 0.48 | 0.43 | 0.46 | 0.46 | 0.47 | 0.49 | 0.54 | 0.51 | 0.50 |
| | MAE | 4.79 | 4.82 | 4.81 | 4.65 | 4.70 | 4.65 | 4.61 | 4.64 | 4.65 | 4.64 | 4.67 | 4.64 | 4.78 | 4.63 | 4.71 | 4.83 |
| | nRMSE | 0.47 | 0.48 | 0.48 | 0.46 | 0.46 | 0.46 | 0.46 | 0.46 | 0.46 | 0.45 | 0.46 | 0.45 | 0.46 | 0.44 | 0.45 | 0.46 |
| Silt | RMSE | 14.68 | 14.32 | 16.19 | 14.03 | 13.95 | 13.66 | 13.88 | 13.95 | 13.62 | 14.04 | 13.36 | 13.84 | 13.63 | 13.67 | 13.65 | 13.59 |
| | $R^2$ | 0.30 | 0.34 | 0.16 | 0.36 | 0.36 | 0.39 | 0.36 | 0.39 | 0.42 | 0.37 | 0.45 | 0.42 | 0.41 | 0.41 | 0.44 | 0.50 |
| | MAE | 11.08 | 10.81 | 12.31 | 10.84 | 10.51 | 10.49 | 10.62 | 10.81 | 10.69 | 10.89 | 10.68 | 10.85 | 10.80 | 10.98 | 10.94 | 10.83 |
| | nRMSE | 0.47 | 0.45 | 0.51 | 0.44 | 0.44 | 0.43 | 0.44 | 0.44 | 0.43 | 0.44 | 0.42 | 0.44 | 0.43 | 0.43 | 0.43 | 0.43 |
| EC | RMSE | 9.22 | 9.12 | 11.13 | 8.54 | 8.45 | 8.25 | 8.46 | 7.57 | 7.87 | 8.05 | 7.16 | 7.28 | 7.53 | 7.77 | 7.42 | 8.04 |
| | $R^2$ | 0.39 | 0.41 | 0.17 | 0.40 | 0.42 | 0.34 | 0.48 | 0.47 | 0.46 | 0.53 | 0.64 | 0.50 | 0.54 | 0.59 | 0.59 | 0.47 |
| | MAE | 3.72 | 4.49 | 4.98 | 4.26 | 3.87 | 4.44 | 3.75 | 4.16 | 4.47 | 3.90 | 4.08 | 3.66 | 4.32 | 4.01 | 4.43 | 4.55 |
| | nRMSE | 2.34 | 2.31 | 2.82 | 2.17 | 2.16 | 2.09 | 2.15 | 1.92 | 2.00 | 2.04 | 1.82 | 1.85 | 1.91 | 1.97 | 1.88 | 2.04 |
| pH | RMSE | 0.21 | 0.21 | 0.22 | 0.21 | 0.21 | 0.21 | 0.21 | 0.22 | 0.21 | 0.21 | 0.21 | 0.21 | 0.20 | 0.21 | 0.20 | 0.20 |
| | $R^2$ | 0.12 | 0.11 | 0.06 | 0.10 | 0.18 | 0.13 | 0.13 | 0.20 | 0.18 | 0.20 | 0.21 | 0.23 | 0.30 | 0.25 | 0.31 | 0.38 |
| | MAE | 0.16 | 0.15 | 0.17 | 0.16 | 0.16 | 0.16 | 0.16 | 0.17 | 0.16 | 0.16 | 0.16 | 0.16 | 0.16 | 0.16 | 0.16 | 0.14 |
| | nRMSE | 0.03 | 0.03 | 0.03 | 0.03 | 0.03 | 0.03 | 0.03 | 0.03 | 0.03 | 0.03 | 0.03 | 0.03 | 0.03 | 0.03 | 0.03 | 0.03 |

The results showed that among the best individual models to predict soil properties, the highest and the lowest prediction accuracies were obtained for pH (nRMSE = 0.03) and gypsum (nRMSE = 1.10) using RF and ANN models, respectively (Table 3). Several studies concluded that RF and ANN were also effective in predicting soil properties in the arid and semiarid regions of Iran [8–10].

### 3.4. Comparison of Model Averaging Techniques

This study compared seven model averaging approaches to the individual base learners (Table 3). Among these techniques, GRA showed the highest prediction accuracy for SOM, CCE, Silt, and pH; BMA was most effective at predicting sand and clay contents; and BIC and BGA were most effective in predicting gypsum and EC, respectively (Table 3). BGA and GRA resulted in the least and the most accurate prediction for EC (nRMSE = 1.82) and pH (nRMSE = 0.03), respectively (Table 3). Diks and Vrugt [25] found that the BGA method produced the highest accuracy for hydrologic systems compared with the other model averaging methods (e.g., EWA, BGA, BMA, and MMA).

The results of this study confirmed our original expectations that, compared to the individual base learners, all model averaging techniques resulted in similar or more accurately predicted soil properties [25]. Notably, the success of the model averaging techniques highly depended on having a diverse set of base learners when making a final prediction. This might be one reason for why model averaging techniques were consistently more effective than the base learners regardless of the predicted soil property. Similarly, Malone et al. [71] compared four techniques for model averaging and recommended the GRA approach for DSM applications; furthermore, their study also showed that model averaging could increase the accuracy and robustness of the individual base learners. The effectiveness of model averaging in DSM has subsequently been demonstrated in multiple other studies [72]. Although it was not tested here, the application of stacked generalization techniques using the SuperLearner algorithm has shown that combining the predictions of multiple base learners into an ensemble learner often resulted in similar or better predictions [73]. In Taghizadeh-Merhjardi et al. [73], the SuperLearner and the EWA techniques consistently outperformed 12 base learners when predicting 12 soil properties for the Urmia Lake region of Iran.

### 3.5. Uncertainty Analysis

To assess the uncertainty of the models, the proportion of measured soil values of the validation data that fell within the 90% prediction interval (i.e., prediction interval coverage probability; PICP) and mean prediction interval (MPI: upper prediction limit minus the lower prediction limit) were calculated. Theoretically, 90% of the observations should fall within the defined prediction interval with a confidence level of 90% and MPI should be as narrow as possible. Among the eight soil properties and nine base learners, the ANN model achieved the highest PICP in predicting SOM, CCE, and gypsum content, while the RF model achieved the highest PICP in predicting EC and pH. The best performing model with the lowest uncertainty for particle size classes varied where ANFIS, XGB, and DNN were most effective in predicting the sand, silt, and clay fractions, respectively (Table 4). Furthermore, the uncertainty analysis showed the trend that the model averaging techniques generally produced higher PICP values and that were closer to the nominal 90% for all soil properties in comparison to the base learners. For example, the PICP values of the GRA model were 91% and 86%, respectively, for SOM and CCE. In terms of MPI (Table 5), for eight soil properties and for all ML models, the estimated mean prediction interval for the model averaging techniques were always smaller than those for the base learners. For example, MPI obtained for SOM ranged from 1.0 to 1.4% for the base learners, while it ranged from 0.7 to 0.9% for the model averaging techniques. This further indicated that the model averaging techniques decreased the uncertainty of the models for predicting soil properties. Notably, there was some uncertainty in the predicted values that may have been related to the high variability in soil properties; low precision of predictions; the inherently poor relationships between soil properties and covariates; and errors in modeling.

**Table 4.** Uncertainty of the models for predicting soil properties (prediction interval coverage probability; PICP).

| Soil Property | Base Learner (%) | | | | | | | | | Model Averaging Technique (%) | | | | | | |
|---|---|---|---|---|---|---|---|---|---|---|---|---|---|---|---|---|
| | kNN | SVR | GP | Lasso | ANN | DNN | ANFIS | RF | XGB | EWA | BGA | AIC | BIC | BMA | MMA | GRA |
| SOM | 26 | 32 | 33 | 36 | 57 | 40 | 42 | 45 | 43 | 66 | 67 | 67 | 72 | 77 | 87 | 91 |
| CCE | 38 | 38 | 35 | 61 | 63 | 40 | 44 | 51 | 57 | 75 | 71 | 77 | 81 | 78 | 81 | 86 |
| Gypsum | 35 | 29 | 23 | 28 | 59 | 50 | 28 | 54 | 55 | 63 | 82 | 65 | 90 | 84 | 82 | 85 |
| Sand | 50 | 43 | 35 | 42 | 56 | 55 | 52 | 67 | 62 | 73 | 70 | 68 | 86 | 80 | 87 | 83 |
| Clay | 31 | 29 | 26 | 36 | 49 | 47 | 56 | 53 | 56 | 71 | 77 | 68 | 68 | 88 | 86 | 87 |
| Silt | 26 | 39 | 23 | 45 | 52 | 66 | 68 | 62 | 69 | 74 | 90 | 84 | 84 | 87 | 85 | 90 |
| EC | 24 | 26 | 20 | 28 | 29 | 35 | 32 | 42 | 39 | 46 | 65 | 62 | 62 | 51 | 63 | 64 |
| pH | 50 | 52 | 45 | 53 | 53 | 55 | 55 | 71 | 70 | 89 | 89 | 84 | 82 | 81 | 82 | 91 |

**Table 5.** Uncertainty of the models for predicting soil properties (mean prediction interval; MPI).

| Soil Property | Base Learner | | | | | | | | | Model Averaging Technique | | | | | | |
|---|---|---|---|---|---|---|---|---|---|---|---|---|---|---|---|---|
| | kNN | SVR | GP | Lasso | ANN | DNN | ANFIS | RF | XGB | EWA | BGA | AIC | BIC | BMA | MMA | GRA |
| SOM | 1.4 | 1.3 | 1.2 | 1.2 | 0.9 | 1.2 | 1.3 | 1.0 | 1.2 | 0.8 | 0.9 | 0.9 | 0.8 | 0.7 | 0.8 | 0.8 |
| CCE | 87.4 | 77.1 | 80.5 | 80.9 | 65.2 | 67.0 | 84.9 | 68.1 | 72.7 | 55.8 | 50.1 | 57.9 | 53.7 | 57.7 | 45.7 | 46.5 |
| Gypsum | 36.5 | 38.5 | 34.4 | 39.2 | 32.9 | 31.7 | 37.9 | 33.2 | 34.6 | 31.2 | 30.6 | 30.0 | 28.8 | 31.5 | 29.4 | 29.9 |
| Sand | 92.9 | 76.1 | 78.4 | 83.0 | 87.3 | 79.4 | 88.9 | 76.2 | 87.8 | 61.9 | 69.9 | 71.1 | 74.4 | 69.8 | 61.7 | 72.4 |
| Clay | 32.4 | 33.1 | 31.3 | 33.0 | 27.9 | 30.0 | 29.8 | 29.0 | 33.2 | 26.7 | 26.6 | 25.7 | 24.2 | 23.0 | 24.9 | 22.1 |
| Silt | 73.8 | 66.2 | 70.7 | 78.5 | 74.8 | 77.7 | 67.3 | 70.6 | 66.8 | 59.3 | 55.1 | 53.0 | 63.8 | 53.0 | 56.5 | 52.7 |
| EC | 37.5 | 25.0 | 34.3 | 31.2 | 37.0 | 31.4 | 29.2 | 22.2 | 25.7 | 21.3 | 15.3 | 18.3 | 14.0 | 16.9 | 16.0 | 18.4 |
| pH | 0.8 | 0.8 | 0.7 | 0.8 | 0.7 | 0.7 | 0.7 | 0.7 | 0.8 | 0.7 | 0.5 | 0.6 | 0.7 | 0.7 | 0.5 | 0.5 |

*3.6. Spatial Prediction of Soil Properties*

The spatial predictions of the target soil properties are illustrated in Figure 4. Based on a visual assessment, the soil maps were consistent with our expert knowledge of the soil patterns for the region and our understanding of the relationships between soil properties, geology, and climate. As expected, the spatial patterns of the SOM predictions were similar to the mean annual precipitation patterns, where the highest amounts of rainfall occur in the western, northwestern, southwestern, and southern parts of the Isfahan province and at higher elevations and lower temperatures, hence facilitating SOM accumulation.

The spatial variability of lime in the soils of the Isfahan province did not match the climatic patterns and instead followed the geological patterns of the study area. Lower amounts of lime were observed along with a northwestern to southwestern corridor within the study area, where the parent material of soils was derived mainly from volcanic rocks [28]. Similarly, low amounts of lime were also predicted within the western region of the Isfahan province, where the parent materials are derived from metamorphic rocks [28].

Soil salinity and gypsum levels increased along an eastern gradient within the Isfahan province. Due to the higher elevation, the western and southern regions of the study area experience greater humidity; hence, gypsum and other soluble minerals are easily leached from the soil profile. In contrast, the regions that were predicted with the highest amounts of gypsum and other soluble salts (Figure 4) corresponded to the regions with lower precipitation and higher temperature, which provide climatic conditions that are conducive for evaporation and, as a result, the formation of gypsum and other soluble salts. Gypsum is often found in soils with calcite and other soluble salts [28]. The parent material types and evaporation were the main factors for the accumulation of gypsum, calcite, and other soluble salts in the arid and semiarid regions [74]. However, it should be highlighted that the mechanism of salinization is quite complicated and may be affected by other factors [75,76]; for example, the accumulation of salt on the soil surface and soil profile may be significantly affected by the spatiotemporal dynamics of soil water content [77,78].

The variability in soil pH was limited, with predicted values ranging between 7.35 and 8.33 and with soil pH being the lowest in the western and southern regions of the province. Similar to the soil salinity and gypsum predictions, we believe that the spatial pattern of soil pH was partially controlled by the climate, where the higher precipitation levels led to the leaching of soluble minerals, thereby decreasing the pH (Figure 4).

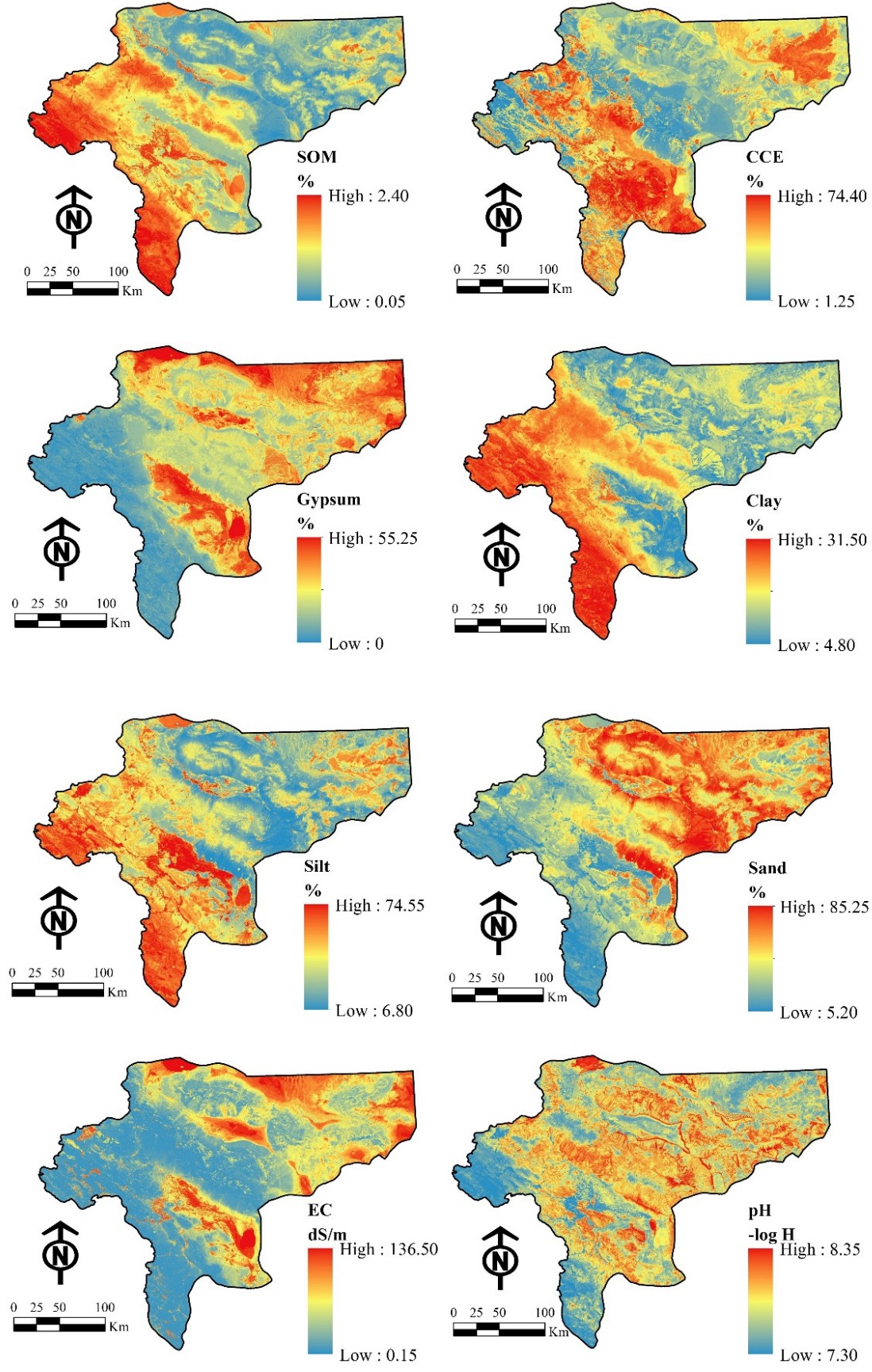

**Figure 4.** Spatial distribution soil properties for the Isfahan province at a 30 m spatial resolution. The maps shown are produced using the best-performing model averaging technique.

Clay and silt contents were highest in the western, southwestern, and southern parts of the Isfahan province, while the opposite trends were observed for sand contents (Figure 4). It appears that both parent material and climate were effective predictors of particle size fractions. In the central, southeastern, northern, and eastern parts of the province, sand dunes, Quaternary sediments, andesite, granite, and diorite were the dominant parent materials from which the resulting soils formed from these parent materials would have a correspondingly high sand content. Furthermore, wind erosion in the eastern region of the province causes an increased loss of finer soil particles and increased sand contents. In the western, southwestern, and southern regions of the Isfahan province, the dominant parent materials consist of sedimentary rocks, such as marls, limestone, shale, and sandstones, thus resulting in soils with higher clay and silt contents. Furthermore, the higher moisture in these regions facilitates higher weathering rates on the soil parent materials, contributing to clay and silt particle production.

## 4. Conclusions

This study evaluated multiple base learners and model averaging approaches for predicting the spatial distribution of soil properties in central Iran. We concluded that among the base learners, the ANN and RF models were the most consistent in predicting soil properties and had a higher accuracy than the other base learners. Furthermore, when comparing the model averaging techniques against the individual base learners, model averaging consistently performed better than the best performing base learner regardless of model averaging techniques and soil property. This might be related to the fact that the model averaging techniques combined the strengths of the base learners in order to obtain a better predictive performance and make the ensemble models more robust than their constituents. Specifically, the GRA and BMA approaches performed the best for all soil properties. The uncertainty analysis showed similar trends in the ML models for predicting soil properties where the model averaging methods had higher PICP and lower MPI values than the base learners. The resulting maps, produced at a 30 m spatial resolution, can be used as valuable baseline information for the effective management of environmental resources. These maps will support the sustainable management of the region's soil resources and facilitate land evaluation activities.

**Author Contributions:** Conceptualization, R.T.-M. and F.K.; methodology, investigation and formal analysis, R.T-M., F.K. and M.Z.; software, R.T.-M.; validation, R.T.-M. and F.K.; resources, H.K. and F.K.; data curation, H.K. and F.K.; writing—original draft preparation, R.T.-M., H.K., F.K., M.Z., B.H. and T.S.; writing—review and editing, R.T.-M., H.K., F.K., M.Z., B.H. and T.S.; visualization, R.T.-M.; supervision, H.K. and T.S.; project administration, H.K. and T.S.; funding acquisition, H.K. and F.K. All authors have read and agreed to the published version of the manuscript.

**Funding:** This research received no external funding.

**Data Availability Statement:** The data presented in this study are available on request from the corresponding author. The data are not publicly available due to privacy restrictions.

**Acknowledgments:** R.T.-M. and T.S. thank the German Research Foundation (DFG) for supporting this research through the Collaborative Research Center (SFB 1070) 'ResourceCultures' (subprojects Z, S, and B02) and the DFG Cluster of Excellence "Machine Learning—New Perspectives for Science", EXC 2064/1, project number 390727645. We acknowledge support from the Open Access Publishing Fund of the University of Tübingen.

**Conflicts of Interest:** The authors declare no conflict of interest.

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
