# Peer review of "A Comparison of Model Averaging Techniques to Predict the Spatial Distribution of Soil Properties"

_remotesensing, doi:10.3390/rs14030472_

Round 1

Reviewer 1 Report

In this paper, the authors presented a study to predict the spatial distribution of soil properties. In fact, they tested and evaluated a suite of nine individual machine learning and seven model averaging techniques to predict the spatial distribution of calcium carbonate equivalent (CCE), soil organic matter (SOM) content, soil pH, gypsum content, electrical conductivity (EC), and particle size fractions for the Isfahan province of central Iran.
For experimental results, the proposed approach has been validated on real remote sensing datasets.
The proposed idea is interesting, however, some revisions have to be made and some parts of these experiments are not complete to claim the advantage of the proposed idea :

1)    The contributions of this paper seem to be not clear to the reviewer. Please clarify the differences between the proposed idea and some existing methods based on machine/deep learning (GCN, CNN, Autoencoders,  etc). For example, which contributions are existing and which ones are your own?
2)    In experimental results, the authors randomly selected the training samples. What happens when you change the selected samples (another random sampling of training data)?
3)     The authors claimed that the model averaging approaches outperform the state-of-the-art methods. How do they explain the high performance of these approaches?) 
4) I suggest the authors to add these recent references which are related to remote sensing analysis  based on deep learning and spatial feature extraction :
- Deep neural networks-based relevant latent representation learning for hyperspectral image classification, Pattern Recognition, 2022.
- A shallow network for hyperspectral image classification using an autoencoder with convolutional neural network, Multimedias tools and applications, 2021.

5) The English and format of this manuscript should be checked very carefully

Author Response

Reviewer 1

In this paper, the authors presented a study to predict the spatial distribution of soil properties. In fact, they tested and evaluated a suite of nine individual machine learning and seven model averaging techniques to predict the spatial distribution of calcium carbonate equivalent (CCE), soil organic matter (SOM) content, soil pH, gypsum content, electrical conductivity (EC), and particle size fractions for the Isfahan province of central Iran. For experimental results, the proposed approach has been validated on real remote sensing datasets. The proposed idea is interesting, however, some revisions have to be made and some parts of these experiments are not complete to claim the advantage of the proposed idea:

We thank the reviewer and accordingly we edited the paper.

The contributions of this paper seem to be not clear to the reviewer. Please clarify the differences between the proposed idea and some existing methods based on machine/deep learning (GCN, CNN, Autoencoders,  etc). For example, which contributions are existing and which ones are your own?

Done, we added some text at the end of Introduction section.

In experimental results, the authors randomly selected the training samples. What happens when you change the selected samples (another random sampling of training data)?

Thanks for raising this issue. We tried to separate the training and testing datasets in a such way that both datasets follow the same distribution and parameters (mean, SD, CV, …). In addition, we calculated, in the current version of the paper, the uncertainty based on the left one out cross-validations (Table 4). This gives us a more guarantee that by changing the training and testing, randomly, the results might not be changed significantly.

The authors claimed that the model averaging approaches outperform the state-of-the-art methods. How do they explain the high performance of these approaches?)

Done, we added some text at the Result section.

I suggest the authors to add these recent references which are related to remote sensing analysis  based on deep learning and spatial feature extraction :

- Deep neural networks-based relevant latent representation learning for hyperspectral image classification, Pattern Recognition, 2022.

- A shallow network for hyperspectral image classification using an autoencoder with a convolutional neural network, Multimedia tools, and applications, 2021.

Done.

The English and format of this manuscript should be checked very carefully

Done, the English was double-checked by Brandon Heung, our native speaker.

Reviewer 2 Report

In this study, the authors tested the ensemble through rescanning the covariate space to maximize the prediction accuracy. The research work is promising. It adds to the body of literatures cited. However, the following points should be considered:

  1. Recent articles available for similar studies can be read.

[1] Wang J, Shi T, Yu D, et al. Ensemble machine-learning-based framework for estimating total nitrogen concentration in water using drone-borne hyperspectral imagery of emergent plants: A case study in an arid oasis, NW China[J]. Environmental Pollution, 2020, 266: 115412.

[2] Peterson K T, Sagan V, Sidike P, et al. Machine learning-based ensemble prediction of water-quality variables using feature-level and decision-level fusion with proximal remote sensing[J]. Photogrammetric Engineering & Remote Sensing, 2019, 85(4): 269-280.

  1. Please provide a bit more big-picture motivation of how aquatic vegetation analyses benefit society and how they have evolved over the past decade. However, from my point of view, the article does not provide a sufficiently thorough review of the issue under study. There are good references for the study techniques, but the paper is missing a "big-picture" introduction with some references in my opinion. I suggest that the authors should do a better analysis of the literature. It seems that the bulk of the text is a sort of compilation of statements in the individual articles cited. It would be better, I think, to extract ideas from individual articles and tie them together into a more fluid and conceptually homogeneous text. As it is, the text looks rather clumsy.
  2. Please explain why model averaging is specifically studied. It is very important.
  3. The soil sampling is not fully described. I would suggest that the authors more clearly articulate the basic information concerning the selection of soil sample sites and present the breakdown on the number of sample sites. While the number of sampling sites points is rather modest, I don't think it is inappropriate considering the large size of the study area and the fairly widely diverging soil properties that occur within this study area. All this being said, I do think the authors should justifying their number of soil sample sites as appropriate for their study.
  4. L101 The superscript and/or subscript through the manuscript should be exceptionally careful. Please double check.
  5. The detailed information of Landsat-8 data (such as capture time, resolution, path/row, and others) should be added. In addition, the corresponding preprocessing steps should be illustrated in the revision.
  6. L135-136 It is hard to follow. Rephrase
  7. L191 normalized root mean squared error (nRMSE). Please revise it.
  8. All used characters in equations should be illustrated in details. Moreover, all variables should be italicized. Please double check.
  9. The version information and corresponding citations of mentioned R-packages should be added.
  10. Currently, the conclusion section is too weak. In fact, some crucial findings which mentioned in abstract should be here.

Author Response

Reviewer 2

In this study, the authors tested the ensemble through rescanning the covariate space to maximize the prediction accuracy. The research work is promising. It adds to the body of literatures cited. However, the following points should be considered:

We thank the reviewer and accordingly we edited the paper.

Recent articles available for similar studies can be read.

[1] Wang J, Shi T, Yu D, et al. Ensemble machine-learning-based framework for estimating total nitrogen concentration in water using drone-borne hyperspectral imagery of emergent plants: A case study in an arid oasis, NW China[J]. Environmental Pollution, 2020, 266: 115412.

[2] Peterson K T, Sagan V, Sidike P, et al. Machine learning-based ensemble prediction of water-quality variables using feature-level and decision-level fusion with proximal remote sensing[J]. Photogrammetric Engineering & Remote Sensing, 2019, 85(4): 269-280.

Great point, these references are used to improve the introduction and result sections.

Please provide a bit more big-picture motivation of how aquatic vegetation analyses benefit society and how they have evolved over the past decade. However, from my point of view, the article does not provide a sufficiently thorough review of the issue under study. There are good references for the study techniques, but the paper is missing a "big-picture" introduction with some references in my opinion. I suggest that the authors should do a better analysis of the literature. It seems that the bulk of the text is a sort of compilation of statements in the individual articles cited. It would be better, I think, to extract ideas from individual articles and tie them together into a more fluid and conceptually homogeneous text. As it is, the text looks rather clumsy.

Great point, we tried to improve the writing the introduction parts.

Please explain why model averaging is specifically studied. It is very important.

Done, we added some text at the end of Introduction section.

The soil sampling is not fully described. I would suggest that the authors more clearly articulate the basic information concerning the selection of soil sample sites and present the breakdown on the number of sample sites. While the number of sampling sites points is rather modest, I don't think it is inappropriate considering the large size of the study area and the fairly widely diverging soil properties that occur within this study area. All this being said, I do think the authors should justifying their number of soil sample sites as appropriate for their study.

Done, we added a full description of sampling in the method section. Although, there are many methods for sampling (design and model based), in the current work, we chose a grid method. This is mainly due to the fact that grid sampling is an appropriate sampling option for mapping. In addition, there were limitations in the budget which force us not only to take a small sample size but also to have a good spatial coverage in the vast area. Noted, we had to ignore a complete structural grid sampling at some points because there were practical constrains in the fields.  

L101 The superscript and/or subscript through the manuscript should be exceptionally careful. Please double check.

Done.

The detailed information of Landsat-8 data (such as capture time, resolution, path/row, and others) should be added. In addition, the corresponding preprocessing steps should be illustrated in the revision.

Done.

L135-136 It is hard to follow. Rephrase

Done.

L191 normalized root mean squared error (nRMSE). Please revise it.

Done.

All used characters in equations should be illustrated in details. Moreover, all variables should be italicized. Please double check.

Done.

The version information and corresponding citations of mentioned R-packages should be added.

Done.

Currently, the conclusion section is too weak. In fact, some crucial findings which mentioned in abstract should be here.

Done.

Round 2

Reviewer 2 Report

Thanks for the great efforts from authors which improved the manuscript a lot. The revised manuscript well addressed most of my previous concerns. The responses to my comments are well prepared and satisfactory. My remaining questions are minor:

  1. The radical motivation and prediction mechanism should be summarized and included in the abstract. Abstract should be modified after whole revising, as a shorter one, and should be clear to appear understandable overall contents (i.e., including objective, method, result, discussion, conclusion), but not too long. Try to make it more expressive and professional.
  2. L374 Common sense: The mechanism of salinization is quite complicated, and this specific phenomenon may be affected by various factors. The accumulation of salt on the soil surface and soil profile is significantly affected by spatiotemporal dynamics of water content. That's why many previous studies choose multiple variables and regression approach to estimate soil salinity. It's not novel story, but could be one of the clues. That's why many previous studies have been discussing about spectral behavior and/or environmental responses with varied soil properties in a desert, wetland, and soil before constructing the models (WANG et al., Assessing toxic metal chromium in the soil in coal mining areas via proximal sensing: Prerequisites for land rehabilitation and sustainable development. Geoderma; Machine learning-based detection of soil salinity in an arid desert region, Northwest China: A comparison between Landsat-8 OLI and Sentinel-2 MSI; Capability of Sentinel-2 MSI data for monitoring and mapping of soil salinity in dry and wet seasons in the Ebinur Lake region, Xinjiang, China). Hence, the authors should revise the discussion.
  3. In addition, please re-write the conclusion in a concise manner (generally in one short paragraph or several key points). The Conclusions section should present a summary of the main findings and new knowledge, but not repeats of what have been discussed in the previous sections.

Author Response

  1. Abstract is improved
  2. The discussion (soil salinity) is improved
  3. Conclusion is improved

English language and style are also improved